# Regulation of Hole Concentration and Mobility and First-Principle Analysis of Mg-Doping in InGaN Grown by MOCVD

**DOI:** 10.3390/ma14185339

**Published:** 2021-09-16

**Authors:** Lian Zhang, Rong Wang, Zhe Liu, Zhe Cheng, Xiaodong Tong, Jianxing Xu, Shiyong Zhang, Yun Zhang, Fengxiang Chen

**Affiliations:** 1Institute of Semiconductors, Chinese Academy of Sciences, Beijing 100083, China; zhanglian07@semi.ac.cn (L.Z.); liuzhe@semi.ac.cn (Z.L.); zhecheng@semi.ac.cn (Z.C.); 2Microsystem and Terahertz Research Center, China Academy of Engineering Physics, Chengdu 610200, China; rong_wang@zju.edu.cn (R.W.); zytxd@foxmail.com (X.T.); xujianxing_mtrc@caep.cn (J.X.); zhangshiyong_mtrc@caep.cn (S.Z.); 3Hangzhou Global Scientific and Technological Innovation Center, Zhejiang University, Hangzhou 311200, China; 4Lishui Zhongke Semiconductor Material Co., Ltd., Lishui 323000, China; 5Department of Physics, School of Science, Wuhan University of Technology, Wuhan 430070, China; phonixchen79@whut.edu.cn

**Keywords:** InGaN, hole, interstitial Mg

## Abstract

This work studied the regulation of hole concentration and mobility in p-InGaN layers grown by metalorganic chemical vapor deposition (MOCVD) under an N-rich environment. By adjusting the growth temperature, the hole concentration can be controlled between 6 × 10^17^/cm^3^ and 3 × 10^19^/cm^3^ with adjustable hole mobility from 3 to 16 cm^2^/V.s. These p-InGaN layers can meet different requirements of devices for hole concentration and mobility. First-principles defect calculations indicate that the p-type doping of InGaN at the N-rich limiting condition mainly originated from Mg substituting In (Mg_In_). In contrast with the compensation of nitrogen vacancy in p-type InGaN grown in a Ga-rich environment, the holes in p-type InGaN grown in an N-rich environment were mainly compensated by interstitial Mg (Mg_i_), which has very low formation energy.

## 1. Introduction

InGaN alloys are attractive semiconductor materials due to their tunable bandgap energy (E_g_) range from 0.65 to 3.4 eV. Thin undoped InGaN layers are widely used in light-emitting diodes (LEDs) and laser diodes (LDs) for lighting, displays and communication [1,2,3]. Due to the high absorption coefficient (~10^5^ cm^−1^) and high radiation resistance [4], thick intrinsic InGaN layers are also prospective candidates for solar cells [5]. In green LDs, thick InGaN layers are used as a waveguiding layer with the inherent benefit of a higher confinement factor [6]. Moreover, the valance band maximum (VBM) of InGaN alloys is higher than that of GaN, which reduces the active energy of Mg and contributes to a high hole concentration in gallium nitrides. p-InGaN layers with thickness beyond 100 nm were reported in solar cells [7] and heterojunction bipolar transistors (HBTs) [8] as the hole-injection and p-type conduction layers.

Many scholars focused on improving the crystal quality of p-type InGaN layers to obtain a large hole concentration and maintain high hole mobility [9,10,11,12]. A hole concentration of 7.7 × 10^17^/cm^3^ was achieved on an Mg-doped In_0.04_Ga_0.96_N grown by molecular beam epitaxy (MBE) [13]. By reducing lattice mismatch, a p-type In_0.18_Ga_0.82_N with a hole concentration up to 3 × 10^19^ cm^−3^ was grown on a GaN substrate by plasma-assisted MBE (PA-MBE) [14]. MOCVD was also used to grow a p-type In_0.22_Ga_0.78_N layer with a hole concentration of 5 × 10^18^ cm^−3^ [15]. However, these reported hole mobilities are limited to less than 10 cm^2^/V.s. Not all devices need to maintain high mobility at a higher hole concentration. For example, devices using p-type layers as hole injection layers, high hole concentration and high mobility are both needed to form large hole current. In addition to these, there are some devices that only need high hole concentration, such as p-type E-mode high electron mobility transistors (HEMTs). High hole concentration is useful to deplete two-dimensional electron gas, resulting in high threshold voltage. However, it does not need high hole mobility, which can lead to a large gate current. It is significant to regulate the hole concentration and mobility within a certain range, so as to meet the needs of different devices.

The main factors affecting hole concentration and mobility in p-type InGaN are as follows. First, V-pits form easily in InGaN layers due to a large lattice mismatch and large thermal expansion coefficient between InN and GaN [16]. V-pits can act as traps of holes, which decreases the hole concentration. In addition, the scattering of V-pits reduces hole mobility [17]. Second, there is a high concentration of background electrons in InGaN layers [18]. Positively charged N vacancies (V_N_) are considered to be the source of the high background-electron concentration because of the low formation energy in Ga-rich growth conditions [19]. However, reaction chambers of MOCVD usually present as an N-rich environment because the V/III ratio is usually in the order of thousands during the growth of gallium nitrides. The formation energy of V_N_ is high in N-rich environments [20]. Other defects that can act as donor impurities to compensate holes in Mg-doped InGaN grown by MOCVD are still uncertain.

In this work, a 100 nm V-pit-free p-InGaN layer on a sapphire substrate was grown by MOCVD. By adjusting the growth temperature, the hole concentration could be controlled between 6 × 10^17^/cm^3^ and 3 × 10^19^/cm^3^ with adjustable hole mobility of 3 to 16 cm^2^/V.s. Mesoscopic defects such as spiral mounds and V-pits originated from the screw dislocation core. First-principles defect calculations were carried out to study defects in an Mg-doped InGaN layer at N-rich limit conditions. The results indicate that the p-type doping of InGaN mainly originated from the process of Mg substituting In (Mg_In_). For p-InGaN grown in the N-rich environment, native defects compensating holes were interstitial Mg (Mg_i_) instead of V_N_ in the p-InGaN grown in the Ga-rich environment.

## 2. Materials and Methods

Under the same pressure of 150 torr, four 100 nm thick samples were grown by MOCVD on 2 inch c-plane sapphire substrates at 790 °C (Sample A), 800 °C (Sample B), 810 °C (Sample C) and 820 °C (Sample D). Triethylgallium (TEGa), trimethylindium (TMIn), bis-cyclopentadienyl-magnesium (Cp_2_Mg) and ammonia (NH_3_) were used as the precursors for Ga, In, Mg and N. Before the InGaN layers, 2 µm thick undoped GaN templates were grown on sapphire. The growth rate of the p-InGaN was about 1 nm per min, and the flux of Cp_2_Mg was 30 sccm for all the samples. Four p-InGaN layers were grown at 820 °C with different growth times of 5, 10, 20 and 50 min. The indium compositions of the InGaN layers were estimated by high-resolution X-ray diffraction (HR-XRD). Surface morphologies of all samples were revealed by atomic force microscopy (AFM, Veeco D3100, New York, NY, USA) with a scan area of 5 × 5 μm^2^. These samples were processed into Vander Pauw geometries for Hall effect measurements at room temperature (RT) in order to test hole concentration and mobility.

## 3. Results and Discussion

### 3.1. p-Type InGaN Growth

Figure 1 shows the ω-2θ scan rocking curves of p-InGaN layers under different growth temperature levels. Indium composition was calculated to be ~7%, 5.8%, 5.3% and 4.8% when the growth temperature was 790, 800, 810 and 820 °C, respectively, according to location of the InGaN peak. When the growth temperature increased, the indium composition was reduced due to the decreased incorporation rate of indium atoms. When the temperature exceeded 800 °C, the indium composition decreased to less than 6%. This is consistent with reported results based on the miscibility rate of InN in GaN, which revealed that the indium content was limited to less than 6% when the growth temperature was higher than 800 °C [21].

Figure 2 displays the AFM images (**a**–**d**) and line scans (**e**–**h**) of the images of the samples with a scan area of 5 × 5 μm^2^. Surface roughness was improved when the growth temperature increased. According to the line scans of the AFM images, the surface fluctuation of Samples A–D shrank. There were many V-pits with a density of ~1.08 × 10^8^/cm^2^ in the morphology of Sample A. In Samples B–D, the V-pits disappeared. In addition, for all the samples, some spiral mounds were observed on the surface.

In order to study the formation mechanism of the mesoscopic V-pits and spiral mound, four p-InGaN layers were grown at 820 °C with different growth times of 5, 10, 20 and 50 min. According to the calibrated growth rate of 1 nm/min by using the scanning electron microscope (SEM, S-4800, made in Japan) of the cross-section of Sample D, we determine that the thickness of the four InGaN layers was 5, 10, 20 and 50 nm, respectively. The morphologies of these layers are shown in Figure 3a–d. The line scans of these AFM images for different InGaN layers are shown in Figure 3e–h. As growth time increased, the surface fluctuation became larger. Several small spiral mounds could be observed on the surface of the p-InGaN under the growth time of 5 min. With increased growth time, the size of the spiral mounds increased, while the number of spiral mounds remained the same. Two spiral mounds are enlarged in Figure 3c. A dislocation core was observed in the spiral mound and the step bend around the dislocation core. During step-flow growth, one end of the step was pinned by screw dislocation. Then, the step curve around the dislocation core evolved into spiral mounds forming [22,23]. V-pits were only observed on the thickest p-InGaN layer, grown for 50 min, which indicated that these V-pits may have originated from threading dislocations (TDs) [24] or stacking faults in the InGaN layer. The TDs or stacking faults were generated by strain relaxation [25] due to the lattice mismatch between InGaN and GaN.

Figure 4 shows the hole concentration and mobility of all samples tested by Hall at RT. The hole concentration reduced from 2.4 × 10^19^/cm^3^ to 1.5 × 10^18^/cm^3^, and further to 6.3 × 10^17^/cm^3^ when the growth temperature increased from 790 to 800 and 810 °C. However, when the growth temperature continued to increase to 820 °C, the hole concentration increased to 2.4 × 10^18^/cm^3^ instead of decreasing. When the growth temperature increased from 790 to 820 °C, the indium content of InGaN alloys reduced, which led to a larger E_g_. According to calculation:E_g_(In_x_Ga_1−x_N) = xE_g_(In) + (1 − x)E_g_(GaN) − bx(1 − x)
where E_g_(InN) = 0.7 eV, E_g_(GaN) = 3.42 eV, b = 1.3 eV, the band differences (ΔE_g_) among the four samples are given. Compared with E_g_ of Sample A, the E_g_ of Sample B increased by 46 meV. The ΔE_g_ between Samples B and C was 19.3 meV, and the ΔE_g_ between Samples C and D was 19.4 meV. The larger E_g_ caused the VBM to shift to a lower level, which increased the active energy of Mg in InGaN. Correspondingly, the hole concentration reduced. Due to the large ΔE_g_ (46 meV) between Samples A and B, the hole concentration dynamically decreased. However, for Sample D, with the highest growth temperature and lowest In content, the hole concentration did not decrease, but increased. Defect density was reduced due to the high temperature and low In content, which resulted the hole concentration being enhanced.

Figure 4 shows that the hole mobility monotonously increased with an increase in temperature. The high hole concentration of Sample A led to low hole mobility, less than 3 cm^2^/V.s, due to serious carrier scattering and defect scattering. The result is consistent with reported data [26]. For Samples B and C, with a relatively low hole concentration and In content, hole mobility was increased thanks to the reduced carrier scattering and defect scattering. The hole mobility of Sample D reached 16 cm^2^/V.s, which is one of the highest values among those previously reported for p-InGaN layers with a hole concentration over 1 × 10^18^/cm^3^.

The four samples have different advantages. Samples A and B with high hole concentration and low mobility are suitable for p-type HEMTs to obtain high Vth with low gate current. Samples C and D with relatively high hole concentration and high mobility are undoubtedly attractive for some devices that need a high hole injection, such as LEDs and NPN-type HBTs.

### 3.2. First-Principles Defect Calculations of p-Type InGaN

In order to investigate the microscopic defect properties of Mg in InGaN, we carried out first-principles defect calculations of Mg in the supercell of In_4_Ga_44_N_48_, which was approximate to the In composition of the above samples. A random In_4_Ga_44_N_48_ alloy was modeled by the SQS approach to determine cation site occupancies [27,28]. The produced atomic structure for the supercell of In_4_Ga_44_N_48_ is shown in Figure 5. First-principles calculations were carried out using projector augmented wave (PAW) pseudopotentials as implemented in the Vienna ab initio Simulation Package [29,30]. The 3D states of Ga and In atoms are included as valence electrons in the pseudopotentials. The plane-wave cut-off energies were set as 400 eV. Structural optimization was performed with the Perdew–Burke–Ernzerhof (PBE) exchange correlation functional. The atomic structures were optimized until changes in total energy were less than 1 × 10^−5^ eV per cell. During the electronic calculations, the screened hybrid density functional of Heyd, Scuseria and Ernzerhof (HSE06) was employed [31]. The fraction of the screened Fock exchange was set as 0.32, which resulted in bandgap energy for In_4_Ga_44_N_48_ of 3.18 eV. The Monkhorst–Pack scheme with a Γ-centered 2 × 2 × 2 special k-points mesh was adopted to sample the reciprocal space of the supercell [32].

In this work, defect configurations of interstitial Mg (Mg_i_), Mg substituting Ga (Mg_Ga_) and Mg substituting In (Mg_In_) were considered. Dominant intrinsic defects such as V_N_, V_In_ and V_Ga_ were considered. Formation energies of Mg and intrinsic defects were calculated by the mixed k-point scheme. The formation energy of defect α at the charge state *q* [$∆H_{f}\;\left(\alpha ,\;q\right)$] is calculated by [33,34]:(1)$$∆H_{f}\;\left(\alpha ,\;q\right)=∆E\;\left(\alpha ,\;q\right)+\sum n_{i}\mu_{i}+{qE}_{F}$$
where $∆E\;\left(\alpha ,\;q\right)=E\left(\alpha ,\;q\right)-E\left(\mathrm{InGaN}\right)+n_{i}E_{i}+qE_{VBM}$, E(InGaN) and E(α, q) are the total energies of the InGaN alloy and InGaN containing defect α with charge state *q*, respectively. $n_{i}$ is the number of atom *i* transferred from the supercell to the reservoir during the formation the defect, $\mu_{i}$ is the chemical potential of the constituent *i* referenced to its elemental solid/gas with energy $E_{i}$, *E_F_* is the Fermi energy referenced to the VBM of In_4_Ga_44_N_48_.

During the Mg doping in InGaN alloys, the accessible values of the chemical potentials $\mu_{i}$ are limited by a series of requirements [35]. Firstly, the values of $\mu_{i}$ are limited to those values that maintain a stable In_4_Ga_44_N_48_ alloy:(2)$$4\mu_{In}+{44\mu }_{\mathrm{Ga}}+48\mu_{N}=∆H_{f}\;\left({\mathrm{In}}_{4}{\mathrm{Ga}}_{44}N_{48}\right)$$
where $∆H_{f}\;\left({\mathrm{In}}_{4}{\mathrm{Ga}}_{44}N_{48}\right)$ is the formation energy of In_4_Ga_44_N_48_ alloy. Secondly, to avoid the precipitation of elemental host phases and the elemental dopant, the values of $\mu_{i}\;$ are limited by:(3)$$\mu_{In\;}\leq \;0,\;\mu_{Ga\;}\leq \;0,\;\mu_{N\;}\leq \;0,\;\mu_{Mg\;}\leq \;0$$Lastly, to avoid the formation of secondary phases, the values of $\mu_{i}$ are limited by:(4)$$\mu_{Ga}+\mu_{N}\;\leq \;∆H_{f}\;\left(\mathrm{GaN}\right),\;\mu_{In}+\mu_{N}\;\leq \;∆H_{f}\;\left(\mathrm{InN}\right),\;3\mu_{Mg}+2\mu_{N}\;\leq \;∆H_{f}\;\left({\mathrm{Mg}}_{2}N_{3}\right)$$
where$\;∆H_{f}\;\left(\mathrm{GaN}\right)$, $∆H_{f}\;\left(\mathrm{InN}\right)$ and $∆H_{f}\;\left({\mathrm{Mg}}_{2}N_{3}\right)$ are formation energies of InN, GaN and Mg_3_N_2_, respectively. The doping of Mg in InGaN alloys was carried out under the N-rich limit, which corresponds to $\mu_{N}=0$. Solving Equations (2)–(4), we obtained the values of μ for In, Ga, N and Mg of −1.3, 0, −0.18 and −1.50 eV, respectively.

The formation energies of Mg and dominant intrinsic defects in InGaN at the N-rich limit are shown in Figure 6. Firstly, the formation energies of V_In_ and Mg_In_ were smaller than those of V_Ga_ and Mg_Ga_, respectively. The chemical potential of In was much smaller than that of Ga at the N-rich limit. It costs less energy for the removal and Mg substitution of In, compared to the case of Ga. Therefore, the p-type doping of InGaN at the N-rich limit mainly originated from Mg_In_. The (0/−) transition level of Mg_Ga_ occurred at 0.14 eV above the VBM, which was lower than that of Mg_Ga_ in GaN [36]. The defect level of Mg_Ga_ is created from the host valence band. Since the VBM of InN was higher than that of GaN [37], alloying of In increased the VBM of GaN and lowered the defect level of Mg_Ga_ in InGaN. The (0/−) transition level of Mg_In_ was 0.16 eV above the VBM, which was slightly deeper than that of Mg_Ga_. This is because the 5*p* orbital energy of In is slightly higher than the 4*p* orbital energy of Ga. Mg_i_ is a deep donor in InGaN, with the (2+/+) and (+/0) transition levels located at 0.40 and 0.17 eV below the conduction band minimum (CBM), respectively. As the Fermi energy of InGaN spreads throughout the bandgap, the charge state of V_N_ changed from 1+ to 0. In GaN, the stable charge states of V_N_ were 3+ and 1+ as the Fermi energy increased. V_N_ created an empty defect level below the CBM and a half-occupied defect level above the VBM of GaN. Because the CBM of InN was lower than that of GaN, alloying of In into GaN lowered the energy of CBM. The empty defect level of V_N_ moved to an energy level higher than the CBM of InGaN. Therefore, the defect level of V_N_ in InGaN was only the half-occupied defect level above the VBM. Due to the N-rich limiting condition, the formation energy of V_N_ was extremely high. The compensation of p-type doping for InGaN at the N-rich limiting condition was the formation of Mg_i_ rather than V_N_. This result indicates that the p-type doping of InGaN grown by MOCVD cannot be enhanced by simply increasing the concentration of the Mg dopant. Once the concentration of Mg was larger than the sum concentration of Mg_In_ and Mg_Ga_ at the growth temperature, the formation of Mg_i_ was not beneficial to the p-type doping of InGaN.

## 4. Conclusions

We grew 100 nm V-pit-free p-InGaN layers on a sapphire substrate by MOCVD. By adjusting the growth temperature, the hole concentration could be controlled between 6 × 10^17^/cm^3^ and 3 × 10^19^/cm^3^ with adjustable hole mobility of 3 to 16 cm^2^/V.s. A p-InGaN layer with a relatively high hole concentration of 2.4 × 10^19^/cm^3^ and low hole mobility of 3 cm^2^/V.s was obtained at 790 °C. It increased the threshold voltage and decreased the gate current for p-type E-HEMTs. A hole concentration of 2.4 × 10^18^/cm^3^ and high hole mobility of 16 cm^2^/V.s were achieved at 820 °C, which are attractive for devices requiring hole injection layers. Mesoscopic defects such as spiral mounds and V-pits originated from the screw dislocation core. First-principles defect calculations indicated that the p-type doping of InGaN in an N-rich limiting condition was mainly caused by Mg_In_. For p-type InGaN grown in an N-rich environment, hole compensation was the formation of Mg_i_ rather than V_N_ in the p-type doping of GaN grown at the Ga-rich environment. Results indicate that, during the p-type doping of InGaN by MOCVD, reducing Mg_i_ is beneficial to increasing the hole concentration.

## Figures and Tables

**Figure 1 materials-14-05339-f001:**
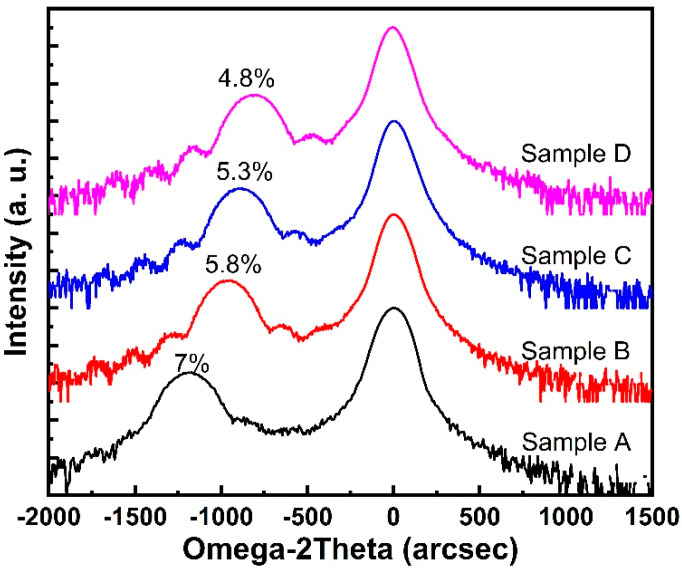
XRD curves of p-InGaN samples (color online).

**Figure 2 materials-14-05339-f002:**
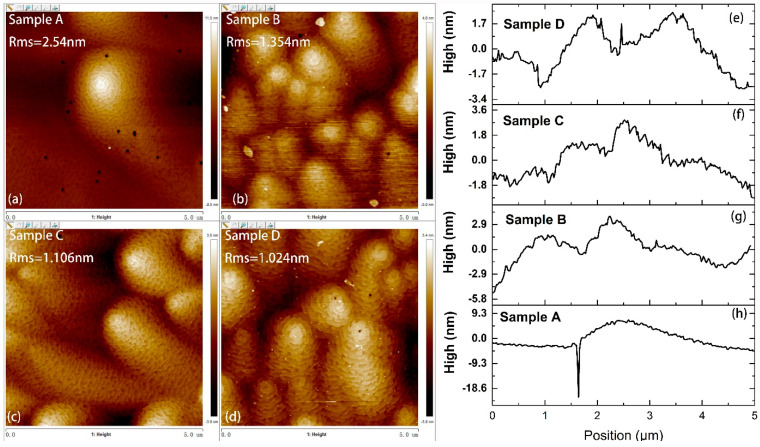
(**a**–**d**) AFM images of p-InGaN samples with scan area of 5 × 5 μm^2^; (**e**–**h**) line scans of AFM images (color online).

**Figure 3 materials-14-05339-f003:**
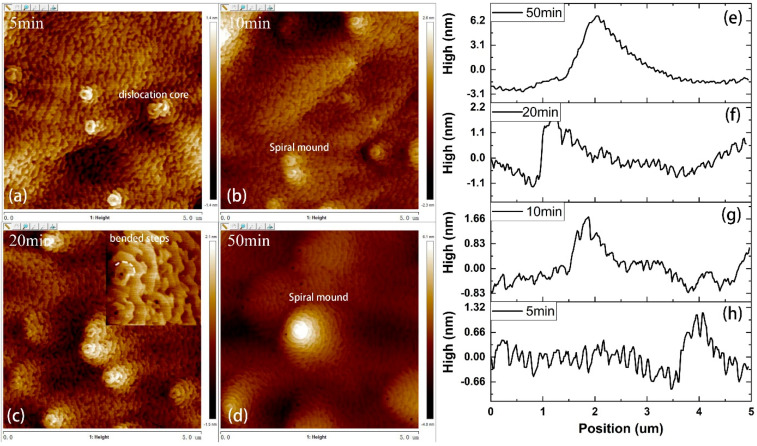
(**a**–**d**) 5 × 5 μm^2^ AFM images of p-InGaN samples with different growth times; (**e**–**h**) line scans of AFM images (color online).

**Figure 4 materials-14-05339-f004:**
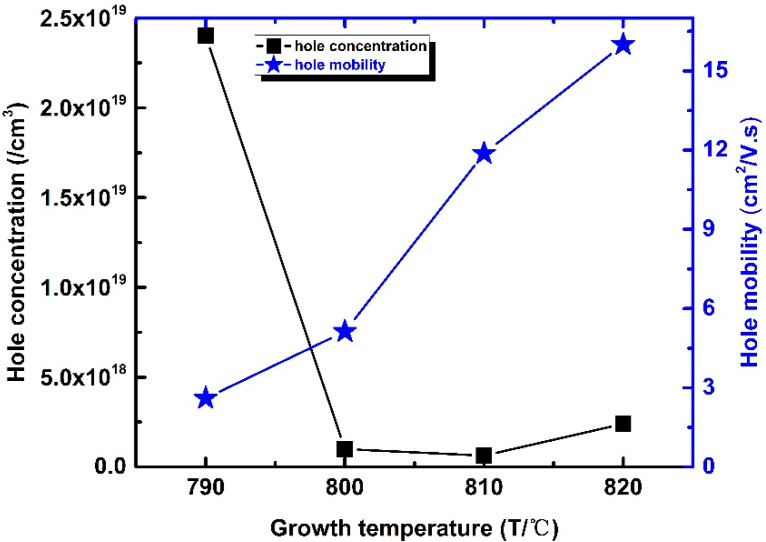
Hole concentration and mobility tested by Hall measurement at RT for the samples (color online).

**Figure 5 materials-14-05339-f005:**
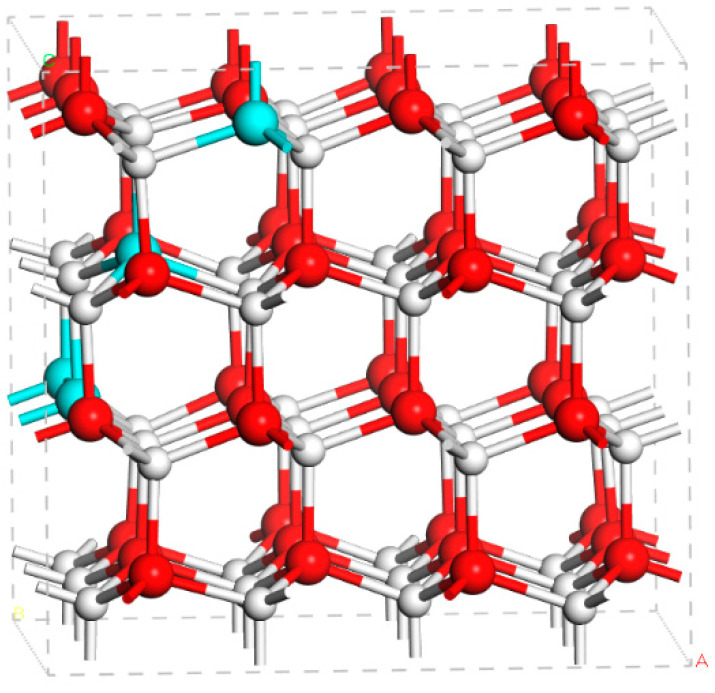
Atomic structure for the supercell of In_4_Ga_44_N_48_. In, Ga and N atoms denoted by cyan, red and white balls, respectively.

**Figure 6 materials-14-05339-f006:**
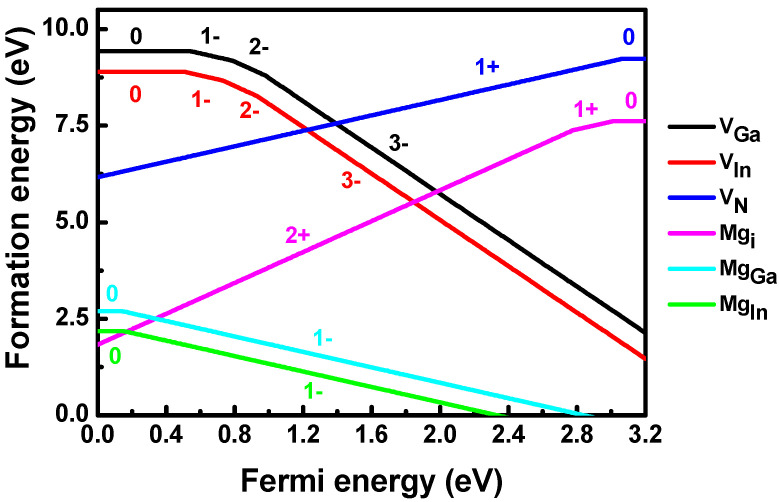
Calculated formation energies of Mg and dominant intrinsic defects as functions of Fermi energy in In_4_Ga_44_N_48_ at the N-rich limit. Stable charge states of each defect are included.

## Data Availability

The data presented in this paper is available with request from the corresponding author.

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
