# Peer review of "Regulation of Hole Concentration and Mobility and First-Principle Analysis of Mg-Doping in InGaN Grown by MOCVD"

_materials, 2021, doi:10.3390/ma14185339_

Round 1

Reviewer 1 Report

The paper "Regulation of hole concentration and mobility and first-princi-2 ple analysis of Mg-doping in InGaN grown by MOCVD" reported about p-type doping effect of MOCVD InGaN by first-principle calculation and supporting experiments, However, the reviewer thinks this paper needs more some discussion for supporting the authors’ opinion. This manuscript can be published in the journal "Materials" after minor revision.

  1. English writing should be modified. I would suggest the author's review again to correct for awkward phrasing and language. And, please double check the typos.
  2. For helping reader’s understanding, the author should modified the definition of sample A, B, C, and D. For example, “Under the same pressure of 150 torr, four 100-nm-thick samples were grown by MOCVD on 2-inch c-plane sapphire substrates at 790℃ (Sample A), 800℃ (Sample B), 810℃ (Sample C) and 820℃ (Sample D)”
  3. In description about Figure 4, the author should explain why hole concentration dynamically decreased from 790 to 800, and why hole concentration incread from 810 to 820.
  4. InGaN has been widely utilized and developed in optoelectronic device. Therefore, The reviewer thinks that the author need to insert some more high impact papers. The reviewer recommends some state-of-arts papers in high-impact journals. In addition to the recommended papers, there are more papers about that, so find and add more.

Advanced Materials 30.28 (2018): 1800649.

Advanced Optical Materials (2021): 2002211.

Author Response

Thanks very much for you recommend. All questions are answered, and major changes in the paper have been marked in red characters. The main comments and our specific responses are detailed in the response letter.

Reviewer 2 Report

The paper of Lian Zhang et al. present how InGaN growth parameters influence on hole concentration and their mobility. Paper is interesting because authors presented not only experimental results of p-InGaN electrical characterization but also tried to explain obtained findings by first-principles calculations. However, current version of the manuscripts needs some corrections which are listed bellow:

  • please uniform carrier concentration notation: /cm3 vs cm-3 (for example line 18 vs 41)
  • In Introduction authors referred to some paper where InGaN layers were investigated and different holes concentration were obtained for different In content. Is there any dependence between indium content in the layer and hole concentration?
  • line 75 - missed subscript in nitrogen vacancy: see line 60 vs 75
  • In part 2. Materials and methods authors mentioned about 4 samples grown at 820°C, please provide information about their thickness.
  • I suppose that all InGaN layers were strained?
  • line 87: Van der Pauw instead of Vander Pauw
  • Were samples grown at 820°C characterized by Hall measurements? Did authors observe thickness dependent changes in hole concentration or its mobility as well as Indium content in the layer?
  • line 92 I would recommend add word rocking in ω-2Θ description
  • what is unit of ω-2Θ in Figure 1 (arc sec?)
  • AFM images (fig 3 and 4) - please provide a line scans for all images to show how the morphology changes. Please also provide a colour scale bar for colour identification
  • To say more about Mg influence on sample morphology it would be nice to compare doped samples with undoped one
  • Please consider some DLTS measurement of InGaN samples. DLTS provide useful information about the defects present in the structure and their activation energy and concentration. The comparison of DLTS result with first-principles calculation will significantly enhance the quality of the paper or will be god ide for another paper

Author Response

Dear reviewer,

I'm glad to receive your valuable comments. Your questions greatly improve the preciseness of this work. According to recommends, some figures are updated, and detailed explanations are given in the revised manuscript. Major revisions have been marked with red characters. The main comments and our specific responses are detailed in the response letter.

Round 2

Reviewer 2 Report

I am satisfied with authors corrections.